# Lower-Left Partial AUC: An Effective and Efficient Optimization Metric for Recommendation

## ABSTRACT

Optimization metrics are crucial for building recommendation systems at scale. However, an effective and efficient metric for practical use remains elusive. While Top-K ranking metrics are the gold standard for optimization, they suffer from significant computational overhead. Alternatively, the more efficient accuracy and AUC metrics often fall short of capturing the true targets of recommendation tasks, leading to suboptimal performance. To overcome this dilemma, we propose a new optimization metric, Lower-Left Partial AUC (LLPAUC), which is computationally efficient like AUC but strongly correlates with Top-K ranking metrics. Compared to AUC, LLPAUC considers only the partial area under the ROC curve in the Lower-Left corner to push the optimization focus on Top-K. We provide theoretical validation of the correlation between LLPAUC and Top-K ranking metrics and demonstrate its robustness to noisy user feedback. We further design an efficient point-wise recommendation loss to maximize LLPAUC and evaluate it on three datasets, validating its effectiveness and robustness. The code is available at https://anonymous.4open.science/r/LLPAUC-D286.

## CCS CONCEPTS

• **Information systems** → **Collaborative filtering**; • **Computing methodologies** → *Machine learning*.

## KEYWORDS

Partial AUC; Recommendation System; Optimization Metrics

**ACM Reference Format:**
Anonymous Author(s). 2018. Lower-Left Partial AUC: An Effective and Efficient Optimization Metric for Recommendation. In *Proceedings of Make sure to enter the correct conference title from your rights confirmation emai (Conference acronym 'XX).* ACM, New York, NY, USA, 14 pages. https://doi.org/XXXXXXX.XXXXXXX

## 1 INTRODUCTION

Recommender systems, core engines for Web applications, aim to alleviate Web information overload by recommending the Top-K most relevant items for each user [30]. They are widely adopted in large-scale Web applications such as Amazon and TikTok [4], and typically learned from historical user feedback using optimization metrics related to item ranking [23]. While Top-K ranking metrics

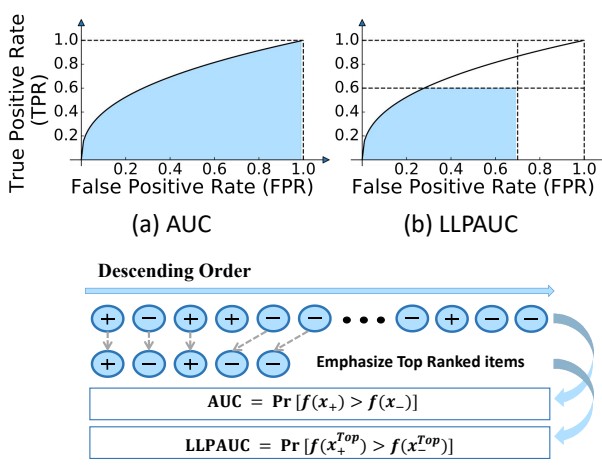

**Figure 1: (a) AUC measures the entire area under the ROC curve; (b) LLPAUC considers the lower-left corner; (c) Compared to AUC, LLPAUC only considers the ranking for top-ranked items.**

such as NDCG@K and Recall@K align well with the goals of recommendation tasks, they are not suitable for practical use at scale due to their substantial computational cost [23]. There thus remains a need to explore effective and efficient optimization metrics for recommender models.

Prior research pursues the target through the trade-off between efficiency and alignment with the Top-K ranking. One approach is to frame the recommendation task as a classification problem and optimize the accuracy metric [5], which inherently deviates from the Top-K ranking. Another approach optimizes the Area Under the Receiver Operating Characteristic (ROC) curve (AUC) metric [24] as shown in Figure 1(a), which quantifies the probability of ranking a random positive item higher than a negative one. AUC accounts for item ranking but treats all items equally, which may not improve the ranking quality for Top-K items when optimized, leading to suboptimal recommendation performance.

In this work, we propose a new optimization metric, Lower-Left Partial AUC, designed to be more correlated with Top-K ranking than the traditional AUC metric. LLPAUC introduces constraints on the upper bound of False Positive Rate (FPR) and True Positive Rate (TPR), *i.e.,* focusing on the partial area under the ROC curve in the Lower-Left corner as depicted in Figure 1(b). These constraints can narrow the ranking to only include the top-ranked items as shown in Figure 1(c), strengthening the correlation with Top-K metrics. Our theoretical analysis shows that LLPAUC can tighter bound Top-K ranking metrics. Notably, the constraint on TPR can also prevent the optimization from overfitting noise user feedback [28], making LLPAUC more robust than AUC.

Nevertheless, the optimization of LLPAUC is non-trivial due to the non-differentiable and computationally expensive TPR and FPR constraint operations. To address these challenges, following [25], we reformulate the constraint operations using the average Top-K loss [7] to make it differentiable and amenable to mini-batch optimization. On top of these efforts, we propose a minimax point-wise loss function, which efficiently maximizes the LLPAUC metric. Moreover, both time complexity analysis and empirical results on real-world datasets verify its efficiency.

The main contributions of the paper are summarized as follows:

- We propose a new optimization metric LLPAUC for recommendation, and provide both theoretical and empirical evidence on its stronger correlation with Top-K ranking metrics.
- We derive an efficient point-wise loss function for maximizing the LLPAUC metric, which has comparable complexity as conventional point-wise recommendation losses.
- We conduct extensive experiments on three datasets under both clean and noisy settings, demonstrating the effectiveness and robustness of optimizing LLPAUC for recommendation.

## 2 RELATED WORK

In this section, we briefly introduce the optimization metrics and loss functions for the recommendation task and review recent studies in partial AUC and its optimization.

### 2.1 Optimization Metrics In Recommendation

In general, there are two common types of loss functions in recommender systems. Point-wise loss functions such as Binary Cross Entropy (BCE) loss [14] cast the recommendation task into a classification problem and optimize the accuracy metric. Pair-wise loss functions such as Bayesian Personalized Ranking (BPR) loss [24] are optimized to maximize the AUC metric. In addition, softmax cross-entropy loss [5] is also widely used to maximize the likelihood estimation of classification. Despite their optimization efficiency, these loss functions have a significant gap with the ideal Top-K ranking metrics.

Beyond these typically employed loss functions, some approaches aim to directly optimize Top-K ranking metrics, such as NDCG@K [23] and Recall@K [22, 27]. However, these methods are computationally expensive and are not suitable for large-scale applications. To tackle this issue, recent studies have proposed the pAp@K metric [3, 15], which combines partial AUC metric and Precision@K metric. The pAp@K metric represents a specific instance of LLPAUC and offers better alignment with Top-K metrics, which lacks theoretical support. On the contrary, our study introduces the more generalized LLPAUC metric and conducts theoretical analyses and simulated experiments to establish the strong relationship between the LLPAUC metric and Top-K metrics.

### 2.2 Partial AUC And Its Optimization

The concept of partial AUC was initially introduced by [18]. In various applications, such as drug discovery and medical diagnosis, only the partial AUC up to a low false positive rate is of interest [20], which motivates the research on One-way Partial AUC (OPAUC). [26] first discusses the correlation between OPAUC and

Top-K metrics for recommendation. Later, [31] argues that a practical classifier must simultaneously have a high TPR and a low FPR. Hence, they propose a new metric named Two-way Partial AUC (TPAUC), which pays attention to the upper-left head region under the ROC curve. Then, [33] first proposes an end-to-end TPAUC optimization framework, which has a profound impact on subsequent work [34]. Nevertheless, TPAUC does not align with the Top-K ranking metrics in the recommendation. The proposed LLPAUC metric exhibits a stronger correlation with Top-K ranking metrics. Beyond that, LLPAUC can additionally alleviate the issue of label noise in recommender systems.

Regarding the optimization of partial AUC, previous works [6, 16, 19, 21] rely on full-batch optimization and the approximation of the Top (Bottom)-K ranking, leading to immeasurable biases and inefficiency. Recently, novel end-to-end mini-batch optimization frameworks have been proposed [33, 35, 36]. These methods can be extended to optimize our proposed LLPAUC metric. In this work, we utilize an unbiased mini-batch optimization scheme [25] due to its superiority in the previous investigation.

## 3 PRELIMINARY

In this section, we present our task formulation and partial AUC formulation for recommendation.

### 3.1 Task Formulation

The primary objective of a recommender is to learn a score function $f(u, i|\theta)$ which is parameterized by $\theta$ and predicts the preference of a user $u \in \mathcal{U}$ on an item $i \in \mathcal{I}$. In this work, we only focus on $f: \mathcal{U} \times \mathcal{I} \to [0, 1]$. For convenience, we use $f_{u,i}$ to denote $f(u, i|\theta)$. This work focuses on the implicit feedback setting [32], where positive interactions contain all items interacted with by $u$ (denoted by $\mathcal{I}_u^+ \subseteq \mathcal{I}$), and negative interactions correspond to all non-interacted items (denoted by $\mathcal{I}_u^- \subseteq \mathcal{I}$). Typically, the learning process is formulated as:

$$\min_{\theta} \frac{1}{|\mathcal{U}|} \sum_{u \in \mathcal{U}} \sum_{i \in \mathcal{I}_u^+} \sum_{j \in \mathcal{I}_u^-} \frac{1}{|\mathcal{I}_u^+| \cdot |\mathcal{I}_u^-|} L(\theta, u, i, j), \quad (1)$$

where $L(\theta, u, i, j)$ denotes the fitting loss for the the positive item $i$ and negative item $j$ of user $u$. The choice of $L(\cdot)$ determines the optimization metrics. For example, the BPR loss [24] can be selected to optimize AUC, while binary cross-entropy loss [5] can be used to optimize accuracy metrics. During serving, the recommender generates a Top-K recommendation list for each user based on the prediction scores. This work aims to develop optimization metrics that are better aligned with the Top-K ranking metrics and can be optimized efficiently.

### 3.2 AUC And Partial AUC

AUC is a widely considered optimization metric in the recommendation, which is defined as the region enclosed by the ROC curve [2], as Figure 1(a) shows. Given a threshold $t$ and a score function $f$, we can define true positive rates (TPR) and false positive rates (FPR) as $\text{TPR}_u(t) = \mathbf{Pr}(f_{u,i} > t | i \in \mathcal{I}_u^+)$ and $\text{FPR}_u(t) = \mathbf{Pr}(f_{u,j} > t | j \in \mathcal{I}_u^-)$, respectively. For a given value $\xi \in [0, 1]$, let $\text{TPR}_u^{-1}(\xi) = \inf\{t \in \mathbb{R}, \text{TPR}_u(t) < \xi\}$ and $\text{FPR}_u^{-1}(\xi) = \inf\{t \in \mathbb{R}, \text{FPR}_u(t) < \xi\}$. Then,

according to Figure 1(a), AUC can be formulated as:

$$\text{AUC} = \frac{1}{|\mathcal{U}|} \sum_{u \in \mathcal{U}} \int_0^1 \text{TPR}_u \left[ \text{FPR}_u^{-1}(\xi) \right] \mathrm{d}\xi. \quad (2)$$

In the recommendation, AUC quantifies the overall ranking quality with consideration of all items in $\mathcal{I}$, and we can reformulate it to a pair-wise ranking form [11] as follows:

$$\text{AUC} = \frac{1}{|\mathcal{U}|} \sum_{u \in \mathcal{U}} \mathbf{Pr}_{i \sim \mathcal{I}_u^+, j \sim \mathcal{I}_u^-} [f_{u,i} > f_{u,j}], \quad (3)$$

where $\mathbf{Pr}_{i \sim \mathcal{I}_u^+, j \sim \mathcal{I}_u^-} [f_{u,i} > f_{u,j}]$ represents the probability that a positive item $i$ is ranked higher than a negative item $j$ for user $u$.

Recently, One-way Partial AUC (OPAUC) [6] is proposed to better measure Top-K recommendation quality. Different from AUC, OPAUC just focuses on the area with FPR $\leq \beta$, which is equivalent to just focusing on pair-wise ranking between positive items and highly scored negative items (with prediction scores in $[\eta_\beta, 1]$, where $\eta_\beta$ satisfies $\mathbf{Pr}_{j \sim \mathcal{I}_u^-} [f_{u,j} \geq \eta_\beta] = \beta$). Formally,

$$\text{OPAUC}(\beta) = \frac{1}{|\mathcal{U}|} \sum_{u \in \mathcal{U}} \mathbf{Pr}_{i \sim \mathcal{I}_u^+, j \sim \mathcal{I}_u^-} [f_{u,i} > f_{u,j}, f_{u,j} \geq \eta_\beta]. \quad (4)$$

Based on the definition, we could write a non-parametric estimator for OPAUC($\beta$) as follows:

$$\widehat{\text{OPAUC}(\beta)} = \frac{1}{|\mathcal{U}|} \sum_{u \in \mathcal{U}} \sum_{i \in \mathcal{I}_u^+} \sum_{j \in \mathcal{I}_u^-} \frac{\mathbb{I}[f_{u,i} > f_{u,j}] \cdot \mathbb{I}[f_{u,j} \geq \eta_\beta]}{n_u^+ \cdot n_u^-}, \quad (5)$$

where $\mathbb{I}(\cdot)$ denotes the indicator function, $n_u^+$ denotes the size of $\mathcal{I}_u^+$, and $n_u^-$ denotes the size of $\mathcal{I}_u^-$.

## 4 WHEN LLPAUC MEETS WITH RECOMMENDER SYSTEM

In this paper, we introduce a novel metric called Lower-Left Partial AUC, which differs from OPAUC by imposing constraints on both FPR and TPR (*i.e.*, TPR $\leq \alpha$, FPR $\leq \beta$) as shown in Figure 1(b). By placing additional constraints on TPR, LLPAUC can more closely approach Top-K metrics and effectively address noisy user feedback issues. We next present the formal definition of LLPAUC and subsequently provide theoretical and empirical analyses to demonstrate its effectiveness in aligning with Top-K metrics.

• **LLPAUC Definition.** LLPAUC($\alpha,\beta$), as illustrated in Figure 1(b), is defined as *the area of the ROC space that lies below the ROC curve with TPR $\leq \alpha$ and FPR $\leq \beta$*. Similarly to OPAUC, for each user $u$, the constraint TPR $\leq \alpha$ implies only considering positive items with prediction scores in $[\eta_\alpha, 1]$, where $\eta_\alpha$ satisfies that $\mathbf{Pr}_{i \sim \mathcal{I}_u^+} [f_{u,i} \geq \eta_\alpha] = \alpha$. The constraint FPR $\leq \beta$ means considering only negative items with prediction scores in $[\eta_\beta, 1]$, where $\eta_\beta$ satisfies that $\mathbf{Pr}_{j \sim \mathcal{I}_u^-} [f_{u,j} \geq \eta_\beta] = \beta$. These constraints will make LLPAUC focus on measuring the ranking quality between such highly scored positive items and negative items, and we can accordingly formulate LLPAUC($\alpha,\beta$) for model $f$ as:

$$\text{LLPAUC}(\alpha, \beta) = \frac{1}{|\mathcal{U}|} \sum_{u \in \mathcal{U}} \mathbf{Pr}_{i \sim \mathcal{I}_u^+, j \sim \mathcal{I}_u^-} [f_{u,i} > f_{u,j}, f_{u,i} \geq \eta_\alpha, f_{u,j} \geq \eta_\beta]. \quad (6)$$

We can also formulate it in an empirical form as follows:

$$\widehat{\text{LLPAUC}(\alpha, \beta)} = \frac{1}{|\mathcal{U}|} \sum_{u \in \mathcal{U}} \sum_{i \in \mathcal{I}_u^+} \sum_{j \in \mathcal{I}_u^-}$$
$$\frac{\mathbb{I}\left[ f_{u,i} > f_{u,j} \right] \cdot \mathbb{I}\left[ f_{u,i} \geq \eta_\alpha \right] \cdot \mathbb{I}\left[ f_{u,j} \geq \eta_\beta \right]}{n_u^+ \cdot n_u^-}. \quad (7)$$

It is apparent that both AUC and OPAUC are special instances of our proposed LLPAUC metric. Specifically, we have AUC=OPAUC(1,1) and OPAUC($\beta$) = LLPAUC(1,$\beta$).

### 4.1 Theoretical Analysis

In this subsection, we present theoretical evidence that LLPAUC($\alpha,\beta$) is highly correlated with Top-K metrics such as Recall@K and Precision@K when $\alpha$ and $\beta$ are appropriately set.

**THEOREM 1.** *Suppose there are $n^+$ positive items and $n^-$ negative items, where $n^+ > K$ and $n^- > K$. Ranking all items in descending order according to the prediction scores obtained from any model f, we have*

$$\frac{1}{n^+} \lfloor \mathcal{G}_{lower}(LLPAUC(\alpha,\beta)) \rfloor \leq$$
$$\text{Recall@}K \leq \frac{1}{n^+} \lceil \mathcal{G}_{higher}(LLPAUC(\alpha,\beta)) \rceil, \quad (8)$$

$$\frac{1}{K} \lfloor \mathcal{G}_{lower}(LLPAUC(\alpha,\beta)) \rfloor \leq$$
$$\text{Precision@}K \leq \frac{1}{K} \lceil \mathcal{G}_{higher}(LLPAUC(\alpha,\beta)) \rceil, \quad (9)$$

*where $\alpha = \frac{K}{n^+}, \beta = \frac{K}{n^-}$, and*

$$\mathcal{G}_{lower}(LLPAUC(\alpha,\beta)) = K - \sqrt{K^2 - n^+ n^- \times LLPAUC(\alpha,\beta)},$$
$$\mathcal{G}_{higher}(LLPAUC(\alpha,\beta)) = \sqrt{n^+ n^- \times LLPAUC(\alpha,\beta)}. \quad (10)$$

**THEOREM 2.** *The bounds for Top-K metrics in Eq. (8) and Eq. (9) are tighter than the bounds obtained with OPAUC in Theorem 3 of [26].*

The proof of Theorem 1 and 2 can be found in Appendix A and B, respectively. Based on the two theorems, we conclude that:
• LLPAUC($\alpha,\beta$) exhibits a stronger correlation with Top-$K$ metrics like Precision@$K$ and Recall@$K$, when compared to OPAUC($\beta$) and AUC. Therefore, optimizing LLPAUC is expected to yield superior performance in the Top-$K$ metrics.
• In the derived bounds, both $\alpha = \frac{K}{n^+}$ and $\beta = \frac{K}{n^-}$ decrease as $K$ decreases. This implies that while manipulating the value of $K$, adjustments to $\alpha$ and $\beta$ should be made in order to maintain a robust correlation between LLPAUC and the corresponding Top-K metrics.

### 4.2 Empirical Analysis

We now provide empirical evidence to further substantiate the strong correlation between LLPAUC and Top-K metrics. We perform Monte Carlo sampling experiments via simulation. Specifically, we assume that there are $n^+$ positive items and $n^-$ negative items, and take each possible permutation of all items to represent a possible ranking list. We randomly sample 10,000 permutations and

calculate the Pearson correlation coefficient between LLPAUC($\alpha$, $\beta$) and Recall@K with different $\alpha$, $\beta$, and $K$. It should be noted that the trend is consistent across simulations with different numbers of positive and negative samples ($n^+$ and $n^-$). Therefore, without loss of generality, we set $n^+ = 1000$ and $n^- = 50000$, where $\alpha$ and $\beta$ are logarithmically scaled. It is worth noting that the correlations between Recall@K and OPAUC($\beta$) (or AUC) can be observed by examining LLPAUC(1,$\beta$) (or LLPAUC(1,1)). From the Figure 2, we observe that:

(1) The maximum correlation coefficient is obtained when $\alpha < 1$ and $\beta < 1$, with a value exceeding 0.8. This observation provides empirical evidence supporting the proposition that LLPAUC($\alpha$, $\beta$) exhibits a stronger correlation with Top-K metrics compared to OPAUC and AUC metrics, thus validating Theorem 2.

(2) As $K$ decreases, the point that corresponds to the maximum correlation coefficient shifts towards smaller values of $\alpha$ and $\beta$. This aligns with the conclusion drawn from the conditions $\alpha = \frac{K}{n^+}$ and $\beta = \frac{K}{n^-}$ in the bounds of Eq. (8), further reinforcing the validity of our Theorem 1.

Furthermore, we observe that using both $\alpha$ and $\beta$ to regulate TPR or FPR could enhance the alignment of LLPAUC with the Top-K ranking. Additionally, utilizing $\alpha$ to regulate TPR can also increase the robustness against noise, which we next discuss.

• **LLPAUC Enhancing Robustness Against Noise.** As stated in [28], noise-positive interactions are harder to fit in the early training stage for the recommendation, which results in relatively larger losses (lower predicted score) of noise interactions. This phenomenon is also confirmed in our experiments detailed in Appendix H.1. As aforementioned, the constraint TPR$\leq \alpha$ implies LLPAUC only considers positive items with prediction scores $f_{u,i} \geq \eta_\alpha$. In this way, lots of noise-positive interactions are filtered out, which makes LLPAUC enhance model robustness against noise.

## 5 METHOD

In this section, we first introduce the loss function that enables efficient optimization of LLPAUC. We then describe the learning algorithm and discuss its time complexity.

### 5.1 Loss Function

To optimize LLPAUC during model learning, it is necessary to further convert the LLPAUC($\alpha$,$\beta$) in Eq. (7) to a loss function that can be efficiently optimized. This involves transforming the non-differentiable and computationally expensive terms in Eq. (7), including the pair-wise ranking term ($\mathbb{I}\left[f_{u,i} > f_{u,j}\right]$) and TPR and FPR constraint terms ($\mathbb{I}\left[f_{u,i} \geq \eta_\alpha\right]$ and $\mathbb{I}\left[f_{u,j} \geq \eta_\beta\right]$), into low-complexity point-wise loss functions. To this end, we replace the pair-wise ranking term with a decouplable surrogate loss and design an *Average Top-K Trick* inspired by [25] to transform the constraint terms. Specifically, we follow the four steps to derive our loss:

• **Step 1: replacing** $\mathbb{I}\left[f_{u,i} > f_{u,j}\right]$ **with surrogate loss function.** The non-continuous and non-differentiable $\mathbb{I}[f_{u,i} > f_{u,j}]$ in Eq. (7) is also appeared in AUC and OPAUC formulation. To convert it, we adopt an approach similar to that used for AUC and OPAUC, which involves replacing it with a continuous surrogate loss $\ell(f_{u,i} - f_{u,j})$. Under the assumptions below, the surrogate $\ell(\cdot)$ is consistent for LLPAUC maximization [8].

**ASSUMPTION 1.** *We assume $\ell(\cdot)$ is a convex, differentiable and monotonically decreasing function when $\ell(\cdot) > 0$, and $\ell'(0) < 0$.*

Then, maximizing LLPAUC($\alpha$, $\beta$) in Eq. (7) is equivalent to minimizing the following loss:

$$\min_\theta \frac{1}{|\mathcal{U}|} \sum_{u \in \mathcal{U}} \sum_{i \in \mathcal{I}_u^+} \sum_{j \in \mathcal{I}_u^-} \frac{\ell(f_{u,i} - f_{u,j}) \cdot \mathbb{I}\left[f_{u,i} \geq \eta_\alpha\right] \cdot \mathbb{I}\left[f_{u,j} \geq \eta_\beta\right]}{n_u^+ \cdot n_u^-}. \quad (11)$$

• **Step 2: decoupling pair-wise loss into point-wise loss.** By setting $\ell(x) = (1 - x)^2$, a square loss satisfying Assumption 1, we could decouple the total loss into positive and negative item components, resulting in a point-wise loss.

**LEMMA 1.** *(Proof in Appendix C) With $\ell(x) = (1 - x)^2$, the LLPAUC($\alpha$, $\beta$) optimization problem in Eq. (11) is equal to*

$$\min_{\theta,(a,b)\in[0,1]^2} \max_{\gamma \in [-1,1]} \frac{1}{|\mathcal{U}|} \sum_{u \in \mathcal{U}} \sum_{i \in \mathcal{I}_u^+} \frac{\ell_+(f_{u,i})\mathbb{I}\left[f_{u,i} \geq \eta_\alpha\right]}{n_u^+}$$
$$+ \sum_{j \in \mathcal{I}_u^-} \frac{\ell_-(f_{u,j})\mathbb{I}\left[f_{u,j} \geq \eta_\beta\right]}{n_u^-} - \gamma^2, \quad (12)$$

*where $a, b$ and $\gamma$ are learnable parameters, $\ell_+(f_{u,i}) = (f_{u,i} - a)^2 - 2(1 + \gamma)f_{u,i}$, and $\ell_-(f_{u,j}) = (f_{u,j} - b)^2 + 2(1 + \gamma)f_{u,j}$.*

• **Step 3: reformulating TPR and FPR constraint terms using an average top-K trick.** The constraint terms $\mathbb{I}\left[f_{u,i} \geq \eta_\alpha\right]$ and $\mathbb{I}\left[f_{u,j} \geq \eta_\beta\right]$ require selecting highly scored positive and negative items, which renders the loss in Eq. (12) still non-differentiable and difficult to optimize. Fortunately, under certain conditions, $\ell_+(f_{u,i})$ is a monotonic decreasing function *w.r.t* $f_{u,i}$ and $\ell_-(f_{u,j})$ is a monotonic increasing function *w.r.t* $f_{u,j}$, as proven in Appendix D. Then, we could make the item selection process differentiable using the average Top-K reformulation trick introduced below.

**LEMMA 2.** *(Proof in Appendix E) Suppose $\ell_+(f_{u,i})$ is monotonic decreasing w.r.t $f_{u,i}$ and $\ell_-(f_{u,j})$ is monotonic increasing w.r.t $f_{u,j}$, then we have*

$$\sum_{i \in \mathcal{I}_u^+} \left[\ell_+(f_{u,i}) \cdot \mathbb{I}[f_{u,i} \geq \eta_\alpha]\right] = \max_{s^+ \in \mathbb{R}} \sum_{i \in \mathcal{I}_u^+} \left[-\alpha s^+ - [-\ell_+(f_{u,i}) - s^+]_+\right],$$
$$\sum_{j \in \mathcal{I}_u^-} \left[\ell_-(f_{u,j}) \cdot \mathbb{I}[f_{u,j} \geq \eta_\beta]\right] = \min_{s^- \in \mathbb{R}} \sum_{j \in \mathcal{I}_u^-} \left[\beta s^- + [\ell_-(f_{u,j}) - s^-]_+\right],$$

*where $s^+$ and $s^-$ are learnable parameters, and $[x]_+ = max(0, x)$.*

By leveraging the average Top-K reformulation trick presented in the lemma, we can reformulate the LLPAUC optimization problem in Eq. (12) as follows:

$$\min_{\theta,(a,b)\in[0,1]^2} \max_{\gamma \in \Omega_\gamma} \frac{1}{|\mathcal{U}|} \sum_{u \in \mathcal{U}} \{\max_{s^+ \in \mathbb{R}} \sum_{i \in \mathcal{I}_u^+} \frac{-\alpha s^+ - [-\ell_+(f_{u,i}) - s^+]_+}{n_u^+}$$
$$+ \min_{s^- \in \mathbb{R}} \sum_{j \in \mathcal{I}_u^-} \frac{\beta s^- + [\ell_-(f_{u,j}) - s^-]_+}{n_u^-} - \gamma^2\}, \quad (13)$$

where $\Omega_\gamma = [\max(-a, b - 1), 1]$.

• **Step 4: swapping min-max operations.** Solving Eq. (13) directly is challenging since it involves a complicated min-max-min sub-problem (it also contains a manageable min-max-max sub-problem). However, as done in [25], we could swap the order of

Figure 2: Pearson correlation coefficient between Recall@K and LLPAUC$(\alpha, \beta)$.

the latter $\max_\gamma$ and $\min_{s^-}$ operations for the min-max-min sub-problem after applying two preprocessing steps: 1) replacing the non-smooth function $[\cdot]_+$ with the softplus function [9] and 2) adding an $L_2$ regularizer to make Eq. (13) strongly-concave w.r.t. $\gamma$. Finally, according to the min-max theorem [1], we could merge the consecutive min (or max) operations, converting the overall optimization problem into a min-max form. Formally, Eq. (13) could be reformulated as (see Appendix F for the proof):

$$\min_{\{\theta, (a,b) \in [0,1]^2, s^- \in \mathbb{R}\}} \max_{\{\gamma \in \Omega_\gamma, s^+ \in \mathbb{R}\}} \frac{1}{|\mathcal{U}|} \sum_{u \in \mathcal{U}} \{$$
$$\sum_{i \in \mathcal{I}_u^+} \frac{-\alpha s^+ - r_\kappa(-\ell_+(f_{u,i}) - s^+)}{n_u^+}$$
$$+ \sum_{j \in \mathcal{I}_u^-} \frac{\beta s^- + r_\kappa(\ell_-(f_{u,j}) - s^-)}{n_u^-} - (w+1)\gamma^2\}, \quad (14)$$

where $\Omega_\gamma = [\max(-a, b-1), 1]$, and $r_\kappa$ denotes the softplus function. Formally, $r_\kappa(x) = \frac{1}{\kappa} \log(1 + \exp(\kappa \cdot x))$, where $\kappa$ is a hyperparameter. It is easy to show that $r_\kappa(x) \xrightarrow{\kappa \to \infty} [x]_+$, which leads to asymptotically unbiased optimization.

**Remark.** *Our final loss function in Eq. (14) is similar to the one proposed in [25]. However, it is important to emphasize that the primary contribution of our work is not the introduction of a completely new optimization scheme. Rather, our main contribution lies in extending existing optimization methods to align with our novel LLPAUC metric while addressing challenges associated with the coexistence of minima and maxima optimizations.*

• **Learning Algorithm and Time Complexity Analysis.** To solve the above minimax optimization in Eq. (14), we employ a stochastic gradient descent ascent (SGDA) method. The detailed algorithm can be found in Algorithm 1. Concretely, after each update of the gradient, we clip the parameters to ensure that they are within the constraints of the domain. Based on it, we derive that the total per-iteration complexity of our method is the same as classical loss functions such as BPR [24] and BCE [5]. The detailed derivation process and empirical analysis can be found in Appendix G.1 and Appendix G.2, respectively.

---

**Algorithm 1** Stochastic Gradient Descent Ascent Algorithm

1: **Input**: User set $\mathcal{U}$, Item set $\mathcal{I}$, learning parameters $\{\theta, a, b, s^+, s^-, \gamma\}$
2: **Initialize**: Randomly select $\{\theta, a, b, s^+, s^-, \gamma\}$. Let $\tau = \{\theta, a, b, s^-\}$, $\tau' = \{\gamma, s^+\}$
3: **for** $t = 0, 1, \cdots, T$ **do**
4:     Sample a mini-batch positive interaction $\mathcal{B}^+$
5:     Uniformly sample a mini-batch $\mathcal{B}_u^- \in \mathcal{I}_u^-$ for each $(u, i) \in \mathcal{B}^+$.
6:     Compute $\mathcal{F}(\tau, \tau')$ defined in Eq.(14).
7:     Update $\tau_{t+1} = \tau_t - \eta \cdot \nabla_\tau \mathcal{F}(\tau, \tau')$;
8:     Update $\tau'_{t+1} = \tau'_t + \eta \cdot \nabla_{\tau'} \mathcal{F}(\tau, \tau')$;
9:     Update $\tau_{t+1} = \text{Clip}(\tau_{t+1})$;
10:    Update $\tau'_{t+1} = \text{Clip}(\tau'_{t+1})$;
11: **end for**
12: **Return** $\theta_{T+1}$

---

Table 1: The statistics of datasets.

| Dataset | User | Item | Interactions | Sparsity |
|---------|------|------|--------------|----------|
| Adressa_clean | 87,417 | 2,222 | 201,128 | 99.89% |
| Adressa_noise | 122,578 | 3,371 | 201,128 | 99.95% |
| Yelp_clean | 45,542 | 56,876 | 1,752,118 | 99.93% |
| Yelp_noise | 45,549 | 57,268 | 1,752,118 | 99.93% |
| Amazon_clean | 80,452 | 98,649 | 3,113,576 | 99.96% |
| Amazon_noise | 80,458 | 98,657 | 3,113,576 | 99.96% |

## 6 EXPERIMENTS

In this section, we conduct a series of experiments on three datasets to evaluate the effectiveness and robustness of our proposed optimization metric LLPAUC along with the loss function.

### 6.1 Experiments Setting

**Dateset.** We conduct experiments on three real-world datasets: Adressa, Yelp, and Amazon-book. Our dataset selection was made intentionally to cover a broad range of recommendation scenarios

**Table 2: Performance comparison on three datasets with clean training. The best results are highlighted in bold.**

| | Method | Adressa | | Yelp | | Amazon | |
|---|---|---|---|---|---|---|---|
| | | Recall@20 | NDCG@20 | Recall@20 | NDCG@20 | Recall@20 | NDCG@20 |
| MF | BCE | $0.1573_{\pm0.0251}$ | $0.0793_{\pm0.0181}$ | $0.0814_{\pm0.0004}$ | $0.0448_{\pm0.0005}$ | $0.0663_{\pm0.0006}$ | $0.0363_{\pm0.0002}$ |
| | BPR | $0.1800_{\pm0.0204}$ | $0.0991_{\pm0.0144}$ | $0.0647_{\pm0.0005}$ | $0.0358_{\pm0.0002}$ | $0.0695_{\pm0.0001}$ | $0.0384_{\pm0.0007}$ |
| | SCE | $0.2001_{\pm0.0031}$ | $0.1057_{\pm0.0015}$ | $0.0762_{\pm0.0007}$ | $0.0425_{\pm0.0003}$ | $0.0894_{\pm0.0012}$ | $0.0507_{\pm0.0009}$ |
| | CCL | $0.1956_{\pm0.0110}$ | $0.0911_{\pm0.0028}$ | $0.0842_{\pm0.0002}$ | $0.0486_{\pm0.0000}$ | $0.0944_{\pm0.0001}$ | $0.0551_{\pm0.0008}$ |
| | DNS($M$, $N$) | $0.1877_{\pm0.0025}$ | $0.0965_{\pm0.0010}$ | $0.0856_{\pm0.0005}$ | $0.0489_{\pm0.0002}$ | $0.1012_{\pm0.0006}$ | $0.0580_{\pm0.0003}$ |
| | Softmax_v($\rho$, $N$) | $0.1849_{\pm0.0105}$ | $0.0949_{\pm0.0088}$ | $0.0824_{\pm0.0008}$ | $0.0470_{\pm0.0004}$ | $0.1024_{\pm0.0001}$ | $0.0592_{\pm0.0001}$ |
| | LLPAUC | $\mathbf{0.2166_{\pm0.0022}}$ | $\mathbf{0.1214_{\pm0.0009}}$ | $\mathbf{0.0884_{\pm0.0005}}$ | $\mathbf{0.0505_{\pm0.0003}}$ | $\mathbf{0.1076_{\pm0.0007}}$ | $\mathbf{0.0612_{\pm0.0004}}$ |
| | Method | Adressa | | Yelp | | Amazon | |
| | | Recall@20 | NDCG@20 | Recall@20 | NDCG@20 | Recall@20 | NDCG@20 |
| LightGCN | BCE | $0.1897_{\pm0.0004}$ | $0.0935_{\pm0.0002}$ | $0.0905_{\pm0.0003}$ | $0.0517_{\pm0.0004}$ | $0.1149_{\pm0.0003}$ | $0.0660_{\pm0.0003}$ |
| | BPR | $0.1737_{\pm0.0006}$ | $0.0923_{\pm0.0004}$ | $0.0802_{\pm0.0005}$ | $0.0453_{\pm0.0003}$ | $0.0922_{\pm0.0002}$ | $0.0520_{\pm0.0001}$ |
| | SCE | $0.1729_{\pm0.0008}$ | $0.0960_{\pm0.0007}$ | $0.0890_{\pm0.0005}$ | $0.0506_{\pm0.0004}$ | $0.1115_{\pm0.0004}$ | $0.0640_{\pm0.0002}$ |
| | CCL | $0.1926_{\pm0.0008}$ | $0.1014_{\pm0.0009}$ | $0.0915_{\pm0.0006}$ | $0.0528_{\pm0.0005}$ | $0.1007_{\pm0.0000}$ | $0.0614_{\pm0.0001}$ |
| | DNS($M$, $N$) | $0.1830_{\pm0.0035}$ | $0.0952_{\pm0.0006}$ | $0.0962_{\pm0.0003}$ | $0.0550_{\pm0.0002}$ | $0.1056_{\pm0.0004}$ | $0.0597_{\pm0.0002}$ |
| | Softmax_v($\rho$, $N$) | $0.1923_{\pm0.0107}$ | $0.1056_{\pm0.0117}$ | $0.0975_{\pm0.0001}$ | $0.0567_{\pm0.0000}$ | $0.1128_{\pm0.0007}$ | $0.0724_{\pm0.0006}$ |
| | LLPAUC | $\mathbf{0.2311_{\pm0.0004}}$ | $\mathbf{0.1312_{\pm0.0002}}$ | $\mathbf{0.1002_{\pm0.0003}}$ | $\mathbf{0.0573_{\pm0.0004}}$ | $\mathbf{0.1201_{\pm0.0003}}$ | $\mathbf{0.0684_{\pm0.0003}}$ |

and accommodate different dataset sizes. **Adressa** is a news reading dataset from Adressavisen [10], where the clicks with dwell time < 10s are thought of as noisy interactions [28]. **Yelp**[1] is a restaurant recommendation dataset with user ratings from one to five. **Amazon-book**[2] is from the Amazon-Review [12] datasets, containing user interaction ratings with extensive books. A rating score below 3 on Yelp and Amazon-book is regarded as a noisy interaction. The statistics of three datasets can be found in Table 1.

**Training Settings.** We employed two training settings, **clean training** and **noise training**, to verify the effectiveness and robustness of our proposed loss. Following [29], clean training filters out noisy user interactions and divides the remaining data into separate training, validation, and testing sets. In contrast, noise training retains the same testing set as clean training yet adds noisy interactions to the training and validation sets. Note that we keep the numbers of noisy training and validation interactions on a similar scale as clean training for a fair comparison.

**Evaluation Protocols.** Following existing studies [14, 24], we adopt the full-ranking evaluation setting, where we calculate the metrics using all negative samples. Meanwhile, we utilize two popular metrics to evaluate models, Recall@K and NDCG@K with $K = 20$, where higher scores indicate better performance.

**Baselines.** We compare our LLPAUC surrogate loss function with the following representative recommender losses. 1) **Bayesian Personalized Ranking (BPR)** [24] loss is a pair-wise loss function, which optimizes the AUC metric. 2) **Binary Cross-Entropy (BCE)** [14] loss optimizes accuracy metric. 3) **Softmax Cross-Entropy (SCE)** [5] loss is widely used for classification problems and maximizes likelihood estimation of classification. 4) **DNS(**$M$,

$N$) and Softmax_v($\rho$, $N$) are state-of-the-art OPAUC-based loss functions for recommendation system. For clean training, recent 5) **Cosine Contrastive Loss (CCL)** [17] is included in the comparison. For noise training, we add strong denoising baselines 6) **RCE** and **TCE** [28] for comparison.

**Parameter Settings.** For a fair comparison, we choose two representative recommender models, Matrix Factorization (MF) and graph neural network model LightGCN [13], as the backbones for all loss functions. All the models are optimized by the Adam optimizer with a learning rate of 0.001 and a batch size of 128. In the training process, we adopt widely used negative sampling trick [17] to improve the training efficiency. The number of negative items for each positive item is set to 100. For the proposed LLPAUC surrogate loss function, we tune $\alpha$ and $\beta$ within the ranges of {0.1, 0.2, 0.3, 0.4, 0.5, 0.6, 0.7, 0.8, 0.9} and { 0.01, 0.02, 0.05, 0.1, 0.2, 0.5, 0.7, 0.9}. All hyperparameter searches are done relying on the validation set.

## 6.2 Main Results

**Clean Training.** Table 2 shows the performance comparison between the LLPAUC surrogate loss function with various baselines under the clean training setting with MF and LightGCN backbones. Several key observations can be made from the results: 1) LLPAUC consistently achieves the best performance across all three datasets with different backbones, outperforming the other loss functions significantly. This demonstrates that LLPAUC strongly correlates with Top-K metrics compared to other optimization metrics, which is consistent with our previous theoretical analysis and independent of the dataset and the backbones. 2) The performance of BPR is noticeably inferior to that of DNS($M$, $N$) and Softmax_v($\rho$, $N$) on all datasets with different backbones. Drawing upon the prior knowledge that OPAUC has a stronger correlation with Top-K compared

**Table 3: Performance comparison on three datasets with noise training. The best results are highlighted in bold.**

|  | Method | Adressa | | Yelp | | Amazon | |
|---|---|---|---|---|---|---|---|
|  |  | Recall@20 | NDCG@20 | Recall@20 | NDCG@20 | Recall@20 | NDCG@20 |
| MF | BCE | $0.1551_{\pm0.0025}$ | $0.0762_{\pm0.0007}$ | $0.0799_{\pm0.0014}$ | $0.0438_{\pm0.0009}$ | $0.0911_{\pm0.0009}$ | $0.0515_{\pm0.0009}$ |
|  | BPR | $0.1666_{\pm0.0215}$ | $0.0880_{\pm0.0139}$ | $0.0626_{\pm0.0014}$ | $0.0341_{\pm0.0009}$ | $0.0663_{\pm0.0008}$ | $0.0363_{\pm0.0006}$ |
|  | SCE | $0.1938_{\pm0.0010}$ | $0.1062_{\pm0.0007}$ | $0.0738_{\pm0.0003}$ | $0.0406_{\pm0.0009}$ | $0.0840_{\pm0.0010}$ | $0.0470_{\pm0.0011}$ |
|  | TCE | $0.1465_{\pm0.0022}$ | $0.0862_{\pm0.0007}$ | $0.0826_{\pm0.0008}$ | $0.0456_{\pm0.0005}$ | $0.0906_{\pm0.0018}$ | $0.0514_{\pm0.0011}$ |
|  | RCE | $0.1617_{\pm0.0329}$ | $0.0819_{\pm0.0221}$ | $0.0818_{\pm0.0009}$ | $0.0452_{\pm0.0005}$ | $0.0965_{\pm0.0017}$ | $0.0549_{\pm0.0015}$ |
|  | DNS$(M, N)$ | $0.1802_{\pm0.0125}$ | $0.0847_{\pm0.0097}$ | $0.0844_{\pm0.0016}$ | $0.0477_{\pm0.0008}$ | $0.0966_{\pm0.0003}$ | $0.0543_{\pm0.0003}$ |
|  | Softmax_v$(\rho, N)$ | $0.1801_{\pm0.0086}$ | $0.0922_{\pm0.0054}$ | $0.0816_{\pm0.00014}$ | $0.0452_{\pm0.0005}$ | $0.0954_{\pm0.0002}$ | $0.0536_{\pm0.0001}$ |
|  | LLPAUC | $\mathbf{0.2127}_{\pm\mathbf{0.0014}}$ | $\mathbf{0.1189}_{\pm\mathbf{0.0009}}$ | $\mathbf{0.0847}_{\pm\mathbf{0.0007}}$ | $\mathbf{0.0481}_{\pm\mathbf{0.0001}}$ | $\mathbf{0.0998}_{\pm\mathbf{0.0008}}$ | $\mathbf{0.0566}_{\pm\mathbf{0.0006}}$ |
|  | Method | Adressa | | Yelp | | Amazon | |
|  |  | Recall@20 | NDCG@20 | Recall@20 | NDCG@20 | Recall@20 | NDCG@20 |
| LightGCN | BCE | $0.1844_{\pm0.0005}$ | $0.0874_{\pm0.0002}$ | $0.0888_{\pm0.0003}$ | $0.0497_{\pm0.0001}$ | $0.1095_{\pm0.0003}$ | $0.0620_{\pm0.0001}$ |
|  | BPR | $0.1661_{\pm0.0007}$ | $0.0914_{\pm0.0006}$ | $0.0800_{\pm0.0005}$ | $0.0448_{\pm0.0002}$ | $0.0884_{\pm0.0005}$ | $0.0492_{\pm0.0002}$ |
|  | SCE | $0.1732_{\pm0.0008}$ | $0.0936_{\pm0.0005}$ | $0.0916_{\pm0.0003}$ | $0.0514_{\pm0.0003}$ | $0.1068_{\pm0.0003}$ | $0.0604_{\pm0.0002}$ |
|  | TCE | $0.2184_{\pm0.0005}$ | $0.1187_{\pm0.0005}$ | $0.0923_{\pm0.0004}$ | $0.0522_{\pm0.0003}$ | $0.1085_{\pm0.0004}$ | $0.0611_{\pm0.0002}$ |
|  | RCE | $0.2204_{\pm0.0007}$ | $0.1219_{\pm0.0007}$ | $0.0941_{\pm0.0006}$ | $0.0536_{\pm0.0008}$ | $0.1126_{\pm0.0004}$ | $0.0639_{\pm0.0005}$ |
|  | DNS$(M, N)$ | $0.1701_{\pm0.0017}$ | $0.0889_{\pm0.0011}$ | $0.0948_{\pm0.0002}$ | $0.0536_{\pm0.0001}$ | $0.1012_{\pm0.0002}$ | $0.0570_{\pm0.0001}$ |
|  | Softmax_v$(\rho, N)$ | $0.1815_{\pm0.0047}$ | $0.0939_{\pm0.0084}$ | $0.0957_{\pm0.0002}$ | $0.0549_{\pm0.0002}$ | $0.1076_{\pm0.0003}$ | $0.0682_{\pm0.0004}$ |
|  | LLPAUC | $\mathbf{0.2228}_{\pm\mathbf{0.0006}}$ | $\mathbf{0.1231}_{\pm\mathbf{0.0005}}$ | $\mathbf{0.0981}_{\pm\mathbf{0.0007}}$ | $\mathbf{0.0558}_{\pm\mathbf{0.0004}}$ | $\mathbf{0.1165}_{\pm\mathbf{0.0007}}$ | $\mathbf{0.0655}_{\pm\mathbf{0.0005}}$ |

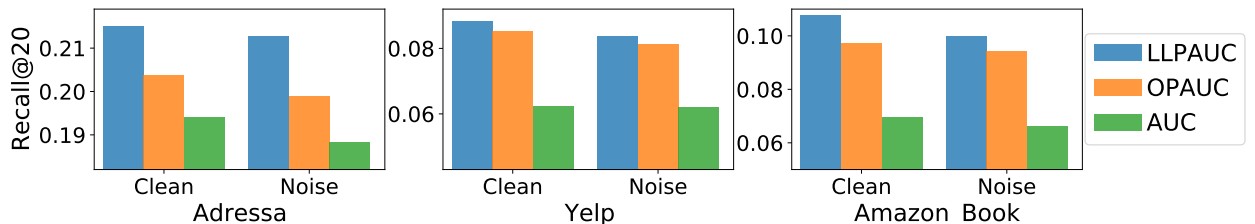

**Figure 3: Ablation studies among different AUC metrics with clean training and noise training.**

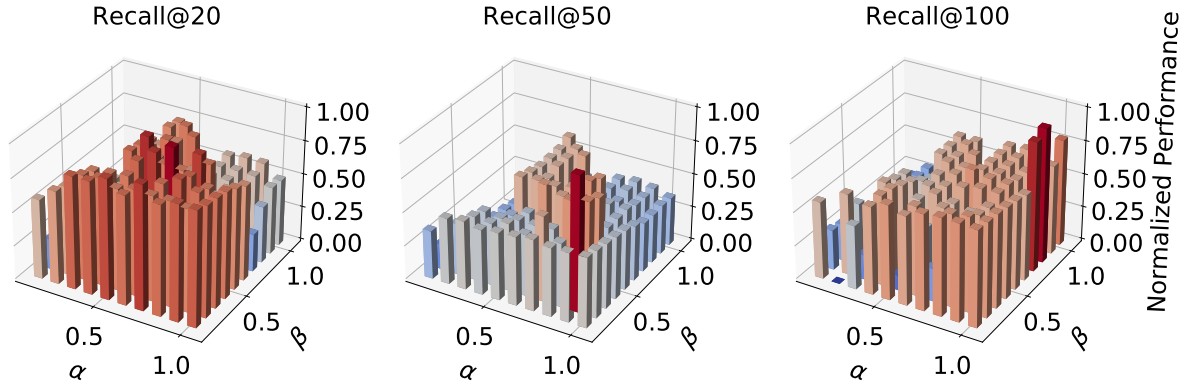

**Figure 4: Normalized Recall@K on Adressa dataset under clean training for K=20, 50 and 100.**

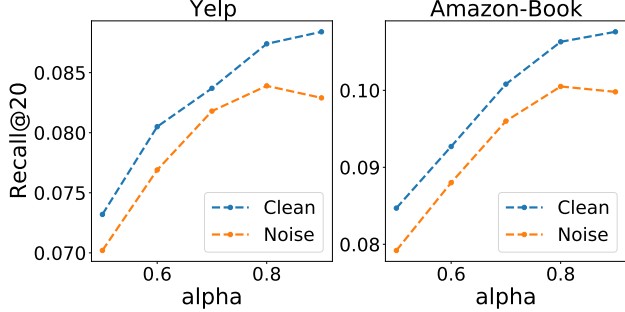

**Figure 5: Given a fix $\beta$, the hyperparameter analysis of $\alpha$ in LLPAUC($\alpha, \beta$) on different datasets under clean training setting and noise training setting.**

to AUC, we can infer that optimization metrics closely tied to Top-K yield superior performance. This finding validates our motivation for proposing LLPAUC. 3) In contrast to BPR and BCE, other losses can implicitly pay more attention to hard negative items, resulting their superior performance. In LLPAUC, we can similarly adjust the attention to hard negative items by varying the $\beta$ parameter. 4) LightGCN outperforms MF in most cases, highlighting its superior strength as a representative graph neural network backbone.

**Noise Training.** In real-world recommender systems, the user interactions collected through implicit feedback often contain natural false-positive interactions. To evaluate the robustness of LL-PAUC, we compare LLPAUC with other loss functions under the noise training setting in Table 3. Notably, we have the following observation: 1) Across all three datasets, the model performance under the noise training setting drops for all loss functions, when compared to the clean training setting. This observation makes sense because it is more challenging to predict user preference from noisy interactions. 2) Denoising baselines like RCE and TCE achieve better performance than other baselines across all datasets, highlighting the importance of noise removal. 3) LLPAUC surpasses all baselines on all datasets, verifying the strong robustness against natural noises. The robustness of LLPAUC stems from its emphasis on higher-ranked positive items, which can be adjusted by hyperparameter $\alpha$.

### 6.3 In-depth Analysis

*6.3.1 Ablation Study.* We next conduct ablation studies to assess the significance of the TPR and FPR constraints in LLPAUC($\alpha, \beta$). Note that restriction on the upper bound of TPR and FPR represents the emphasis on high-ranked positive and negative items in LL-PAUC, respectively. As shown in Eq. (6), OPAUC($\beta$) = LLPAUC($1, \beta$) and AUC = LLPAUC($1, 1$). Based on it, we obtain ablation loss functions of AUC and OPAUC($\beta$) by setting $\alpha$ and $\beta$ in Eq. (14). The results of ablation studies are summarized in Figure 3, where we can observe that: 1) Under clean training, LLPAUC outperforms OPAUC, and OPAUC perform better than AUC. This verifies both emphases on high-ranked positive items and high-ranked negative items strengthen the correlation between LLPAUC and Top-K metrics. 2) When exposed to noisy interactions, LLPAUC demonstrates relatively minor performance degradation compared to OPAUC and

AUC, showcasing its robustness against noise. This is attributed to the emphasis on high-ranked positive items and avoidance of noise samples with low ranks in LLPAUC.

*6.3.2 Hyperparameter Analysis.* To verify the impact of the constraints introduced by LLPAUC for recommender systems, we conduct the grid search experiments on hyperparameters $\alpha$ and $\beta$ and present the corresponding Recall@K performance in Figure 4. To facilitate a better comparison, we report the normalized Recall@K metrics. Concretely, we have Normalized_Recall = $\frac{\text{Recall}-\text{Min\_Recall}}{\text{Max\_Recall}-\text{Min\_Recall}}$. From the figure, we observe that: 1) The maximum performance is obtained with $\alpha < 1$ and $\beta < 1$. Recall that AUC=LLPAUC(1,1)) and OPAUC=LLPAUC(1,$\beta$). Hence, this demonstrates both restrictions of $\alpha$ and $\beta$ of LLPAUC enhance its correlation with the Top-K metric, which is consistent with our Theorem 2 and empirical analysis in Section 4.2. 2) As K in Recall@K decreases, we should shift towards a smaller value of $\alpha$ and $\beta$ to achieve the best performance, empirically corroborating the bound conditions in our Theorem 1. This means we could emphasize different Top-K performances for different K by adjusting $\alpha$ and $\beta$ in LLPAUC.

*6.3.3 Analysis of Robustness.* In this subsection, we conduct experiments to analyze the impact of hyperparameter $\alpha$ on the robustness of the model. Given a fix $\beta$, Figure 5 shows how the LLPAUC model's performance changes *w.r.t* $\alpha$ under clean training and noise training setting. Since the natural noise in the Adressa dataset is relatively weak, we do not include it in our comparison. From the figure, we observe that: 1) Since the noisy interactions impede the model's ability to learn the true interests of users, the performance in the noise training setting consistently falls below that of the clean training setting. This is consistent with our observation in Table 2. 2) Given a fix $\beta$, the maximum Recall@20 performance of LLPAUC is achieved with $\alpha = 0.9$ under clean training settings, and $\alpha = 0.8$ under noisy training settings. This means under the noise training setting, we should choose smaller $\alpha$ to enhance the robustness. Since $\alpha$ constrains TPR in LLPAUC as stated in Eq. (6), we conclude that the emphasis on high-ranked positive items could enhance the model robustness.

## 7 CONCLUSION AND FUTURE WORK

In this work, we presented a novel optimization metric for recommender systems, LLPAUC, to alleviate the dilemma of balancing effectiveness and computational efficiency in previous optimization metrics. In particular, LLPAUC is efficient like AUC while strongly correlating with Top-K ranking metrics, leading to superior Top-K recommendation performance. To optimize LLPAUC, we developed a point-wise loss function and conducted experiments on three datasets, demonstrating its efficiency, effectiveness, and robustness under clean and noise settings.

Future work could shed light on the following limitations of our work: 1) Only focusing on high-ranked positive samples like LLPAUC is not sufficient to fully mitigate the impact of natural noise. 2) The TPR and FPR constraint terms in LLPAUC could be more efficiently reformulated.

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

## A  PROOF OF THEOREM 1

**Reminder of Theorem 1** Suppose there are $n^+$ positive items and $n^-$ negative items, where $n^+ > K$ and $n^- > K$. Ranking all items in descending order according to the prediction scores obtained from any model f, we have

$$\frac{1}{n^+} \lfloor \mathcal{G}_{lower}(\text{LLPAUC}(\alpha, \beta)) \rfloor \leq$$
$$\text{Recall@K} \leq \frac{1}{n^+} \lceil \mathcal{G}_{higher}(\text{LLPAUC}(\alpha, \beta)) \rceil, \quad (15)$$

$$\frac{1}{K} \lfloor \mathcal{G}_{lower}(\text{LLPAUC}(\alpha, \beta)) \rfloor \leq$$
$$\text{Precision@K} \leq \frac{1}{K} \lceil \mathcal{G}_{higher}(\text{LLPAUC}(\alpha, \beta)) \rceil, \quad (16)$$

where $\alpha = \frac{K}{n^+}$, $\beta = \frac{K}{n^-}$, and

$$\mathcal{G}_{lower}(\text{LLPAUC}(\alpha, \beta)) = K - \sqrt{K^2 - n^+ n^- \times \text{LLPAUC}(\alpha, \beta)},$$
$$\mathcal{G}_{higher}(\text{LLPAUC}(\alpha, \beta)) = \sqrt{n^+ n^- \times \text{LLPAUC}(\alpha, \beta)}. \quad (17)$$

PROOF. For any given model $f$, we suppose there are $i(i < K)$ positive items among the Top-K items ranked according to $f$. Then we have Recall@K $= i/n^+$. Under this condition, easily, we can find out the case which has the maximum value of LLPAUC$(\alpha, \beta)$, where $\alpha = \frac{K}{n^+}$ and $\beta = \frac{K}{n^-}$:

$$\underbrace{+ \cdots +}_{i} \underbrace{- \cdots -}_{K-i} | \underbrace{+ \cdots +}_{K-i} \underbrace{- \cdots -}_{i} .$$

Hence, as stated in Eq. (7), the maximum value of LLPAUC$(\alpha, \beta)$ is $\frac{-i^2 + 2Ki}{n^+ n^-}$. Given this, we can deduce the maximum value of Recall@K when LLPAUC$(\alpha, \beta)$ takes a certain value. Note that $i$ can only be integers, we derive that:

$$\frac{1}{n^+} \left\lfloor K - \sqrt{K^2 - n^+ n^- \times \text{LLPAUC}(\alpha, \beta)} \right\rfloor \leq \text{Recall@K}.$$

Similarly, the case that has the minimum value of LLPAUC$(\alpha, \beta)$ is :

$$\underbrace{- \cdots -}_{K-i} \underbrace{+ \cdots +}_{i} | \underbrace{- \cdots -}_{i} \underbrace{+ \cdots +}_{K-i} .$$

Based on Eq. (7), the minimum value of LLAUC$(\alpha, \beta)$ is $\frac{i^2}{n^+ n^-}$. Similarly, we derive the minimum value of Recall@K when LLPAUC$(\alpha, \beta)$ takes a certain value:

$$\text{Recall@K} \leq \frac{1}{n^+} \left\lceil \sqrt{n^+ n^- \times \text{LLPAUC}(\alpha, \beta)} \right\rceil .$$

These complete the proof of Eq. (8). Noticing that for a given permutation, Precision@K $= \frac{n^+}{K} \cdot$ Recall@K, where $\frac{n^+}{K}$ is a constant. Hence, we can easily derive the Eq. (9). □

## B  PROOF OF THEOREM 2

**Reminder of Theorem 2** The bounds for Top-K metrics in Eq. (8) and Eq. (9) are tighter than the bounds obtained with OPAUC in Theorem 3 of [26].

PROOF. Note that the bounds obtained with OPAUC$(\beta)$ in [26] is:

$$\frac{1}{n^+} \lfloor \mathcal{H}_{lower}(\text{OPAUC}(\beta)) \rfloor \leq$$
$$\text{Recall@K} \leq \frac{1}{n^+} \lceil \mathcal{H}_{higher}(\text{OPAUC}(\beta)) \rceil,, \quad (18)$$

$$\frac{1}{K} \lfloor \mathcal{H}_{lower}(\text{OPAUC}(\beta)) \rfloor \leq$$
$$\text{Precision@K} \leq \frac{1}{K} \lceil \mathcal{H}_{higher}(\text{OPAUC}(\beta)) \rceil, \quad (19)$$

where $\beta = \frac{K}{n^-}$ and

$$\mathcal{H}_{lower}(\text{OPAUC}(\beta)) = \frac{n^+ + K - \sqrt{(n^+ + K)^2 - 4n^+ n^- \times \text{OPAUC}(\beta)}}{2},$$
$$\mathcal{H}_{higher}(\text{OPAUC}(\beta)) = \sqrt{n^+ n^- \times \text{OPAUC}(\beta)}. \quad (20)$$

Without loss of generality, we consider the bounds of Recall@K first. To prove that Eq. (8) is a tighter bound than Eq. (18), we need prove that $\mathcal{H}_{lower}(\text{OPAUC}(\beta)) \leq \mathcal{G}_{lower}(\text{LLPAUC}(\alpha, \beta))$ and $\mathcal{H}_{higher}(\text{OPAUC}(\beta)) \geq \mathcal{G}_{higher}(\text{LLPAUC}(\alpha, \beta))$.

**Step 1: Proof of** $\mathcal{H}_{higher}(\textbf{OPAUC}(\beta)) \geq \mathcal{G}_{higher}(\textbf{LLPAUC}(\alpha, \beta))$
For any ranking list ranked by model $f$, we calculate LLPAUC$(\alpha, \beta)$ and OPAUC$(\beta)$ as following:

$$\text{LLPAUC}(\alpha, \beta) = \sum_{i \in \mathcal{I}_u^+} \sum_{j \in \mathcal{I}_u^-} \frac{\mathbb{I}\left[f_{u,i} > f_{u,j}\right] \cdot \mathbb{I}\left[f_{u,i} \geq \eta_\alpha\right] \cdot \mathbb{I}\left[f_{u,j} \geq \eta_\beta\right]}{n^+ \cdot n^-},$$

$$\text{OPAUC}(\beta) = \sum_{i \in \mathcal{I}_u^+} \sum_{j \in \mathcal{I}_u^-} \frac{\mathbb{I}\left[f_{u,i} > f_{u,j}\right] \cdot \mathbb{I}\left[f_{u,j} \geq \eta_\beta\right]}{n^+ \cdot n^-}.$$

where

$$\eta_\alpha = \text{argmin}_{\eta \in \mathbb{R}} \left[ \mathbb{E}_{i \sim \mathcal{I}_u^+}[\mathbb{I}(f_{u,i} \geq \eta)] = \alpha \right],$$

and

$$\eta_\beta = \text{argmin}_{\eta \in \mathbb{R}} \left[ \mathbb{E}_{j \sim \mathcal{I}_u^-}[\mathbb{I}(f_{u,j} \geq \eta)] = \beta \right].$$

When $\alpha = \frac{K}{n^+}$ and $\beta = \frac{K}{n^-}$, LLPAUC$(\alpha, \beta)$ and OPAUC$(\beta)$ can be reformulated as:

$$\text{LLPAUC}(\alpha, \beta) = \sum_{i=1}^{K} \sum_{j=1}^{K} \frac{\mathbb{I}\left[f_{u,[i]} > f_{u,[j]}\right]}{n^+ n^-},$$

$$\text{OPAUC}(\beta) = \sum_{i=1}^{n_u^+} \sum_{j=1}^{K} \frac{\mathbb{I}\left[f_{u,i} > f_{u,[j]}\right]}{n^+ n^-},$$

where $f_{u,[i]}$ denotes the $i$-th largest score among positive items and $f_{u,[j]}$ denotes the $j$-th largest score among negative items. This means LLPAUC$(\alpha, \beta)$ only considers K positive items with the largest prediction scores and K negative items with the largest prediction scores. And OPAUC$(\beta)$ considers all positive items and K negative items with the largest prediction scores.

We categorize and discuss the possible scenarios of the ranking list. In the first scenario, the number of positive samples appearing in descending order reaches K first:

$$\underbrace{\cdots +}_{K \text{ positive },S \text{ negative}} \Big| \underbrace{\cdots}_{(n^+ - K) \text{ positive},(n^- - S) \text{ negative}} ,$$

where $S < K$. And we could observe that $\text{LLPAUC}(\alpha, \beta) \leq \text{OPAUC}(\beta)$. When we keep $\text{LLPAUC}(\alpha, \beta)$ fixed, the maximum value of $\text{OPAUC}(\beta)$ can be achieved as following:

$$\underbrace{\cdots +}_{K \text{ positive }, S \text{ negative}} \quad | \quad \underbrace{+ \cdots +}_{(n^+ - K) \text{ positive}} \quad \underbrace{- \cdots -}_{(n^- - S) \text{ negative}} \quad ,$$

Hence, in the first scenario, we can conclude that

$$\text{LLPAUC}(\alpha, \beta) \leq \text{OPAUC}(\beta) \leq \text{LLPAUC}(\alpha, \beta) + \frac{(n^+ - K) \cdot (n^- - S)}{n^+ n^-}. \tag{21}$$

Easily, we could further obtain that

$$\text{LLPAUC}(\alpha, \beta) \geq \frac{K(n^- - S)}{n^+ n^-}. \tag{22}$$

In the second scenario, the number of negative samples appearing in descending order reaches K first:

$$\underbrace{\cdots -}_{S' \text{ positive }, K \text{ negative}} \quad | \quad \underbrace{\cdots}_{(n^+ - S') \text{ positive}, (n^- - K) \text{ negative}} \quad .$$

And we could find that:

$$\text{OPAUC}(\beta) = \text{LLPAUC}(\alpha, \beta). \tag{23}$$

Taking into account the two scenarios discussed above, we can easily conclude that $\text{OPAUC}(\beta) \geq \text{LLPAUC}(\alpha, \beta)$, which results in $\mathcal{H}_{higher}(\text{OPAUC}(\beta)) \geq \mathcal{G}_{higher}(\text{LLPAUC}(\alpha, \beta))$.

**Step 2: Proof of** $\mathcal{H}_{lower}(\textbf{OPAUC}(\beta)) \leq \mathcal{G}_{lower}(\textbf{LLPAUC}(\alpha, \beta))$
First, we have

$$\mathcal{H}_{lower}(\text{OPAUC}(\beta)) - \mathcal{G}_{lower}(\text{LLPAUC}(\alpha, \beta))$$

$$= \frac{n^+ + K - \sqrt{(n^+ + K)^2 - 4n^+ n^- \text{OPAUC}(\beta)}}{2} -$$

$$\qquad\qquad (K - \sqrt{K^2 - n^+ n^- \text{LLPAUC}(\alpha, \beta)})$$

$$= \frac{1}{2} [(n^+ - K) - (\sqrt{(n^+ + K)^2 - 4n^+ n^- \text{OPAUC}(\beta)} -$$

$$\qquad\qquad 2\sqrt{K^2 - n^+ n^- \text{LLPAUC}(\alpha, \beta)})] \tag{24}$$

Similar to Step 1, we consider two scenarios. In the first scenario, using Eq. (21), we have:

$$\mathcal{H}_{lower}(\text{OPAUC}(\beta)) - \mathcal{G}_{lower}(\text{LLPAUC}(\alpha, \beta))$$

$$\leq \frac{1}{2} \{(n^+ - K) + [4K^2 - 4n^+ n^- \text{LLPAUC}(\alpha, \beta)]^{\frac{1}{2}} - [(n^+ - K)^2 +$$

$$4K(n^+ - K) + 4K^2 - 4n^+ n^- \text{LLPAUC}(\alpha, \beta) - 4(n^+ - K) \cdot (n^- - S)]^{\frac{1}{2}}\} \tag{25}$$

It's notable that when $\sqrt{A^2} + \sqrt{C^2} - \sqrt{A^2 + B^2 + C^2} \leq 0$, we have $2\sqrt{A^2 C^2} - B^2 \leq 0$. Hence, when $A^2 = (N^+ - K)^2$, $B^2 = 4K(n^+ - K) - 4(n^+ - K) \cdot (n^- - S)$, $C^2 = 4K^2 - 4n^+ n^- \text{LLPAUC}(\alpha, \beta)$, to prove

$$\mathcal{H}_{lower}(\text{OPAUC}(\beta)) - \mathcal{G}_{lower}(\text{LLPAUC}) \leq 0, \tag{26}$$

we need to prove that

$$2\sqrt{(n^+ - K)^2 \cdot (4K^2 - 4n^+ n^- \text{LLPAUC}(\alpha, \beta))} - 4(n^+ - K)(n^- - S - K) \leq 0. \tag{27}$$

It's equal to prove that:

$$K^2 - n^+ n^- \text{LLPAUC}(\alpha, \beta) - (n^- - S - K)^2 \leq 0. \tag{28}$$

$$\iff n^+ n^- \text{LLPAUC}(\alpha, \beta) \geq -(n^- - S)^2 + 2(n^- - S)K \tag{29}$$

Since we already have $\text{LLPAUC}(\alpha, \beta) \geq \frac{K(n_u^- - S)}{n^+ n^-}$ in Eq. (22), we can easily complete the proof in the first scenario.

For the second scenario, Eq. (24) can be reformulated as

$$\frac{1}{2} \{(n^+ - K) + \sqrt{4K^2 - 4n^+ n^- \text{LLPAUC}(\alpha, \beta)}$$

$$- \sqrt{(n^+ - K)^2 + 4K(n^+ - K) + 4K^2 - 4n^+ n^- \text{LLPAUC}(\alpha, \beta)} \tag{30}$$

Similarly, to prove

$$\mathcal{H}_{lower}(\text{OPAUC}(\beta)) - \mathcal{G}_{lower}(\text{LLPAUC}) \leq 0, \tag{31}$$

we need to prove that

$$2\sqrt{(n^+ - K)^2 \cdot (4K^2 - 4n^+ n^- \text{LLPAUC}(\alpha, \beta))} + 4(n^+ - K)K \leq 0. \tag{32}$$

It's equal to prove that:

$$K^2 - n^+ n^- \text{LLPAUC}(\alpha, \beta) + K^2 \leq 0. \tag{33}$$

$$\iff \text{LLPAUC}(\alpha, \beta) \geq 0 \tag{34}$$

This is trivially true, thus we have completed the proof for the second scenario. □

## C PROOF OF LEMMA 1

**Reminder of Lemma 1** With $\ell(x) = (1 - x)^2$, the $\text{LLPAUC}(\alpha, \beta)$ optimization problem in Eq. (11) is equal to

$$\min_{\theta, (a,b) \in [0,1]^2} \max_{\gamma \in [-1,1]} \frac{1}{|\mathcal{U}|} \sum_{u \in \mathcal{U}} \sum_{i \in \mathcal{I}_u^+} \frac{\ell_+(f_{u,i}) \mathbb{I}\left[f_{u,i} \geq \eta_\alpha(f)\right]}{n_u^+}$$

$$+ \sum_{j \in \mathcal{I}_u^-} \frac{\ell_-(f_{u,j}) \mathbb{I}\left[f_{u,j} \geq \eta_\beta(f)\right]}{n_u^-} - \gamma^2, \tag{35}$$

where $a, b$ and $\gamma$ are learnable parameters, $\ell_+(f_{u,i}) = (f_{u,i} - a)^2 - 2(1 + \gamma)f_{u,i}$, and $\ell_-(f_{u,j}) = (f_{u,j} - b)^2 + 2(1 + \gamma)f_{u,j}$.

PROOF. We extend the proof of Theorem 7 in [25] to LLPAUC. Given $\ell(x) = (1 - x)^2$, we could reformulate Eq. (11) as:

$$\min_{\theta} \frac{1}{|\mathcal{U}|} \sum_{u \in \mathcal{U}} \sum_{i \in \mathcal{I}_u^+} \sum_{j \in \mathcal{I}_u^-} \frac{(1 - f_{u,i} + f_{u,j})^2 \cdot \mathbb{I}\left[f_{u,i} \geq \eta_\alpha\right] \cdot \mathbb{I}\left[f_{u,j} \geq \eta_\beta\right]}{n_u^+ \cdot n_u^-}$$

$$= \min_{\theta} \frac{1}{|\mathcal{U}|} \sum_{u \in \mathcal{U}} \sum_{i \in \mathcal{I}_u^+} \sum_{j \in \mathcal{I}_u^-} \frac{1}{n_u^+ n_u^-} \{\mathbb{I}\left[f_{u,i} \geq \eta_\alpha\right] \cdot \mathbb{I}\left[f_{u,j} \geq \eta_\beta\right]$$

$$+ f_{u,i}^2 \cdot \mathbb{I}\left[f_{u,i} \geq \eta_\alpha\right] + f_{u,j}^2 \cdot \mathbb{I}\left[f_{u,j} \geq \eta_\beta\right] - 2f_{u,i} \mathbb{I}\left[f_{u,i} \geq \eta_\alpha\right]$$

$$+ 2f_{u,j} \mathbb{I}\left[f_{u,j} \geq \eta_\beta\right] - 2f_{u,i} f_{u,j} \mathbb{I}\left[f_{u,i} \geq \eta_\alpha\right] \cdot \mathbb{I}\left[f_{u,j} \geq \eta_\beta\right]\}. \tag{36}$$

Note that

$$\frac{1}{n_u^+} \sum_{i \in \mathcal{I}_u^+} f_{u,i}^2 \cdot \mathbb{I}\left[f_{u,i} \geq \eta_\alpha\right] - \{\frac{1}{n_u^+} \sum_{i \in \mathcal{I}_u^+} f_{u,i} \cdot \mathbb{I}\left[f_{u,i} \geq \eta_\alpha\right]\}^2$$

$$= \min_{a \in [0,1]} \frac{1}{n_u^+} \sum_{i \in \mathcal{I}_u^+} (f_{u,i} - a)^2 \cdot \left[f_{u,i} \geq \eta_\alpha\right], \quad (37)$$

where the optimal value of a is:

$$a^* = \frac{1}{n_u^+} \sum_{i \in \mathcal{I}_u^+} f_{u,i} \mathbb{I}\left[f_{u,i} \geq \eta_\alpha\right]. \quad (38)$$

Likewise,

$$\frac{1}{n_u^-} \sum_{j \in \mathcal{I}_u^-} f_{u,j}^2 \cdot \mathbb{I}\left[f_{u,j} \geq \eta_\beta\right] - \{\frac{1}{n_u^-} \sum_{j \in \mathcal{I}_u^-} f_{u,j} \cdot \mathbb{I}\left[f_{u,j} \geq \eta_\beta\right]\}^2$$

$$= \min_{b \in [0,1]} \frac{1}{n_u^-} \sum_{j \in \mathcal{I}_u^-} (f_{u,j} - b)^2 \cdot \left[f_{u,j} \geq \eta_\beta\right], \quad (39)$$

where the optimal value of b is:

$$b^* = \frac{1}{n_u^-} \sum_{i \in \mathcal{I}_u^-} f_{u,j} \mathbb{I}\left[f_{u,j} \geq \eta_\beta\right]. \quad (40)$$

Then, We can substitute Eq. (37) and Eq. (39) into Eq. (36) to obtain:

$$\min_{\theta, a, b \in [0,1]^2} \frac{1}{|\mathcal{U}|} \sum_{u \in \mathcal{U}} \{\alpha\beta + \frac{1}{n_u^+} \sum_{i \in \mathcal{I}_u^+} (f_{u,i} - a)^2 \cdot \left[f_{u,i} \geq \eta_\alpha\right]$$

$$+ \left[\frac{1}{n_u^+} \sum_{i \in \mathcal{I}_u^+} f_{u,i} \cdot \mathbb{I}\left[f_{u,i} \geq \eta_\alpha\right]\right]^2 + \frac{1}{n_u^-} \sum_{j \in \mathcal{I}_u^-} (f_{u,j} - b)^2 \cdot \left[f_{u,j} \geq \eta_\beta\right]$$

$$+ \left[\frac{1}{n_u^-} \sum_{j \in \mathcal{I}_u^-} f_{u,j} \cdot \mathbb{I}\left[f_{u,j} \geq \eta_\beta\right]\right]^2 - \frac{1}{n_u^+} \sum_{i \in \mathcal{I}_u^+} 2f_{u,i} \mathbb{I}\left[f_{u,i} \geq \eta_\alpha\right]$$

$$+ \frac{1}{n_u^-} \sum_{j \in \mathcal{I}_u^-} 2f_{u,j} \mathbb{I}\left[f_{u,j} \geq \eta_\beta\right]$$

$$- \frac{1}{n_u^+} \sum_{i \in \mathcal{I}_u^+} \frac{1}{n_u^-} \sum_{j \in \mathcal{I}_u^-} 2f_{u,i} f_{u,j} \mathbb{I}\left[f_{u,i} \geq \eta_\alpha\right] \cdot \mathbb{I}\left[f_{u,j} \geq \eta_\beta\right]\}. \quad (41)$$

It's notable that

$$\left[\frac{1}{n_u^+} \sum_{i \in \mathcal{I}_u^+} f_{u,i} \cdot \mathbb{I}\left[f_{u,i} \geq \eta_\alpha\right]\right]^2 + \left[\frac{1}{n_u^-} \sum_{j \in \mathcal{I}_u^-} f_{u,j} \cdot \mathbb{I}\left[f_{u,j} \geq \eta_\beta\right]\right]^2$$

$$- \frac{1}{n_u^+} \sum_{i \in \mathcal{I}_u^+} \frac{1}{n_u^-} \sum_{j \in \mathcal{I}_u^-} 2f_{u,i} f_{u,j} \mathbb{I}\left[f_{u,i} \geq \eta_\alpha\right] \cdot \mathbb{I}\left[f_{u,j} \geq \eta_\beta\right]$$

$$= \left[\frac{1}{n_u^+} \sum_{i \in \mathcal{I}_u^+} f_{u,i} \cdot \mathbb{I}\left[f_{u,i} \geq \eta_\alpha\right] - \frac{1}{n_u^-} \sum_{j \in \mathcal{I}_u^-} f_{u,j} \cdot \mathbb{I}\left[f_{u,j} \geq \eta_\beta\right]\right]^2$$

$$= \max_\gamma \left[2\gamma\left(\frac{1}{n_u^+} \sum_{i \in \mathcal{I}_u^+} f_{u,i} \cdot \mathbb{I}\left[f_{u,i} \geq \eta_\alpha\right] - \frac{1}{n_u^-} \sum_{j \in \mathcal{I}_u^-} f_{u,j} \cdot \mathbb{I}\left[f_{u,j} \geq \eta_\beta\right]\right) - \gamma^2\right], \quad (42)$$

where the maximization is achieved by:

$$\gamma^* = \frac{1}{n_u^+} \sum_{i \in \mathcal{I}_u^+} f_{u,i} \cdot \mathbb{I}\left[f_{u,i} \geq \eta_\alpha\right]. \quad (43)$$

Easily, $\gamma^* = b^* - a^*$. Then we can constrain $\gamma$ with range $[-1, 1]$ and get equivalent formulation of Eq. (11):

$$\min_{\theta, (a,b) \in [0,1]^2} \max_{\gamma \in [-1,1]} \frac{1}{|\mathcal{U}|} \sum_{u \in \mathcal{U}} \sum_{i \in \mathcal{I}_u^+} \frac{\ell_+(f_{u,i})\mathbb{I}\left[f_{u,i} \geq \eta_\alpha(f)\right]}{n_u^+}$$

$$+ \sum_{j \in \mathcal{I}_u^-} \frac{\ell_-(f_{u,j})\mathbb{I}\left[f_{u,j} \geq \eta_\beta(f)\right]}{n_u^-} - \gamma^2, \quad (44)$$

where $a, b$ and $\gamma$ are learnable parameters, $\ell_+(f_{u,i}) = (f_{u,i} - a)^2 - 2(1 + \gamma)f_{u,i}$, and $\ell_-(f_{u,j}) = (f_{u,j} - b)^2 + 2(1 + \gamma)f_{u,j}$. □

## D PROOF OF FUNCTION

In this subsection, we utilize the following lemmas to substantiate our argument.

**LEMMA** 3. *If $\gamma \in [b - 1, 1]$, $\ell_-(f_{u,j}) = (f_{u,j} - b)^2 + 2(1 + \gamma)f_{u,j}$ is an increasing function w.r.t $f_{u,j}$, when $j \in \mathcal{I}_u^-$ and $f_{u,j} \in [0, 1]$.*

The proof can be found in Appendix F.2.2 in [25].

**LEMMA** 4. *If $\gamma \in [\max\{b - 1, -a\}, 1]$, $\ell_+(f_{u,i}) = (f_{u,i} - a)^2 - 2(1 + \gamma)f_{u,i}$ is an increasing function w.r.t $f_{u,i}$, when $i \in \mathcal{I}_u^+$ and $f_{u,i} \in [0, 1]$.*

The proof can be found in Appendix F.3.2 in [25].

## E PROOF OF LEMMA 2

**Reminder of Lemma 2** Suppose $\ell_+(f_{u,i})$ is monotonic decreasing *w.r.t.* $f_{u,i}$ and $\ell_-(f_{u,j})$ is monotonic increasing *w.r.t.* $f_{u,j}$, then we have

$$\sum_{i \in \mathcal{I}_u^+} \left[\ell_+(f_{u,i}) \cdot \mathbb{I}[f_{u,i} \geq \eta_\alpha]\right] = \max_{s^+ \in \mathbb{R}} \sum_{i \in \mathcal{I}_u^+} \left[-\alpha s^+ - \left[-\ell_+(f_{u,i}) - s^+\right]_+\right], \quad (45)$$

$$\sum_{j \in \mathcal{I}_u^-} \left[\ell_-(f_{u,j}) \cdot \mathbb{I}[f_{u,j} \geq \eta_\beta]\right] = \min_{s^- \in \mathbb{R}} \sum_{j \in \mathcal{I}_u^-} \left[\beta s^- + \left[\ell_-(f_{u,j}) - s^-\right]_+\right], \quad (46)$$

where $s^+$ and $s^-$ are learnable parameters, and $[x]_+ = max(0, x)$ for any $x$.

PROOF. For Eq. (46), the proof can be found in Lemma 1 in [7]. To prove Eq. (45), we first denote that $(-\ell_+(f_{u,i}))$ is monotonic increasing *w.r.t* $f_{u,i}$ and then obtain:

$$\sum_{i \in \mathcal{I}_u^+} \left[\ell_+(f_{u,i}) \cdot \mathbb{I}[f_{u,i} \geq \eta_\alpha]\right]$$

$$= - \sum_{i \in \mathcal{I}_u^+} \left[(-\ell_+(f_{u,i})) \cdot \mathbb{I}[f_{u,i} \geq \eta_\alpha]\right]$$

$$= - \min_{s^+ \in \mathbb{R}} \sum_{i \in \mathcal{I}_u^+} \left[\alpha s^+ + \left[-\ell_+(f_{u,i}) - s^+\right]_+\right]$$

$$= \max_{s^+ \in \mathbb{R}} \sum_{i \in \mathcal{I}_u^+} \left[-\alpha s^+ - \left[-\ell_+(f_{u,i}) - s^+\right]_+\right]. \quad (47)$$

This completes our proof of Eq. (45). Notably that in the final line of derivation, we employ $-\min f(x) = \max -f(x)$. □

## F    PROOF OF MIN-MAX SWAP

Proof. To swap $\max_\gamma$ and $\min_{s^-}$, according to the min-max theorem [1], we need to check the second part of Eq. (13) strongly-concave *w.r.t.* $\gamma$. Concretely, the function is:

$$\mathcal{F}_2 = \sum_{j \in \mathcal{I}_u^-} \frac{\beta s^- + \frac{1}{\kappa}(\log(1 + \exp(\kappa \cdot (\ell_-(f_{u,j}) - s^-))))}{n_u^-} - (w+1)\gamma^2, \tag{48}$$

where $\ell_-(f_{u,j}) = (f_{u,j} - b)^2 + 2(1 + \gamma)f_{u,j}$. Hence,

$$\frac{\partial \mathcal{F}_2}{\partial \gamma} = \frac{1}{n_u^-} \sum_{j \in \mathcal{I}_u^-} \frac{\exp(\kappa \cdot (\ell_-(f_{u,j}) - s^-))}{1 + \exp(\kappa \cdot (\ell_-(f_{u,j}) - s^-))} \cdot 2f_{u,j} - 2w\gamma, \tag{49}$$

$$\frac{\partial^2 \mathcal{F}_2}{\partial \gamma^2} = \frac{1}{n_u^-} \sum_{j \in \mathcal{I}_u^-} \frac{\exp(\kappa \cdot (\ell_-(f_{u,j}) - s^-))}{\left[1 + \exp(\kappa \cdot (\ell_-(f_{u,j}) - s^-))\right]^2} \cdot 4\kappa f_{u,j}^2 - 2w. \tag{50}$$

Since $f_{u,j} \in [0, 1]$ and $\frac{\exp(\kappa \cdot (\ell_-(f_{u,j}) - s^-))}{\left[1 + \exp(\kappa \cdot (\ell_-(f_{u,j}) - s^-))\right]^2} \in (0, 1)$, with suffi-ciently large $w > 4\kappa$, we have $\frac{\partial^2 \mathcal{F}_2}{\partial \gamma^2} < 0$. Therefore, with sufficiently large $w$, Eq. (13) is strongly-concave *w.r.t.* $\gamma$. □

## G    METHOD

### G.1    Time Complexity

For time complexity analysis, we need to consider both forward and backward computational complexity. As stated in Eq. (14), the function is:

$$\mathcal{F} = \frac{1}{|\mathcal{U}|} \sum_{u \in \mathcal{U}} \{ \sum_{i \in \mathcal{I}_u^+} \frac{-\alpha s^+ - r_\kappa(-\ell_+(f_{u,i}) - s^+)}{n_u^+}$$

$$+ \sum_{j \in \mathcal{I}_u^-} \frac{\beta s^- + r_\kappa(\ell_-(f_{u,j}) - s^-)}{n_u^-} - (w+1)\gamma^2 \}. \tag{51}$$

Hence, the complexity of forward propagation is $O(|\mathcal{B}^+||\mathcal{B}^-|d^2)$, where $d$ is the embedding size of user and item, $\mathcal{B}^+$ and $\mathcal{B}^+$ is the mini batch size defined in Algorithm 1. For backward propagation, we first derive the gradient of the function $\mathcal{F}$:

$$\frac{\partial \mathcal{F}}{\partial \theta} = \frac{1}{|\mathcal{U}|} \sum_{u \in \mathcal{U}} \{ \frac{1}{n_u^+} \sum_{i \in \mathcal{I}_u^+} \frac{\exp(\kappa \cdot (-\ell_+(f_{u,i}) - s^+))}{1 + \exp(\kappa \cdot (-\ell_+(f_{u,i}) - s^+))} \cdot \frac{\partial \ell_+(f_{u,i})}{\partial \theta}$$

$$+ \frac{1}{n_u^-} \sum_{j \in \mathcal{I}_u^-} \frac{\exp(\kappa \cdot (\ell_-(f_{u,j}) - s^-))}{1 + \exp(\kappa \cdot (\ell_-(f_{u,j}) - s^-))} \cdot \frac{\partial \ell_-(f_{u,j})}{\partial \theta} \}. \tag{52}$$

Easily, we obtain the complexity of Eq. (52) is $O(|\mathcal{B}^+||\mathcal{B}^-|d^2)$. The partial derivatives of the function with respect to other parameters have a similar form and the same computational complexity. Hence, the total complexity per iteration is $O(|\mathcal{B}^+||\mathcal{B}^-|d^2)$, which is the same with other baseline models such as BPR loss and BCE loss.

### G.2    Experiments Analysis

In this subsection, we show the plots of training convergence on three different datasets under clean training setting in Figure 6. We compare the proposed LLPAUC with three representative baselines: BPR, BCE, and SCE. Easily, we could observe our approach shows a comparable or even faster convergence rate compared to the baseline methods.

## H    EXPERIMENTS

The statistics of three public datasets under clean training and noise training are shown in Table 1, which vary in scale and sparsity. Note that we keep the numbers of noise training and validation interactions on a similar scale as clean training for a fair comparison. Therefore, in the noise training setting, the datasets have more users and items compared to clean training but have the same interactions.

### H.1    Noise Interaction Loss Analysis

In this subsection, we analyze the changes in the average loss of noisy interactions and clean interactions *w.r.t* iterations during the training process of BPR model under noisy training settings. As shown in Figure 7, we observe that during the early stages of model training, the mean loss for noisy interactions is consistently greater than the mean loss for clean interactions. Hence, the emphasis on high-ranked positive items could filter out noise interactions, which makes LLPAUC enhance model robustness against noise.

Received 20 February 2007; revised 12 March 2009; accepted 5 June 2009

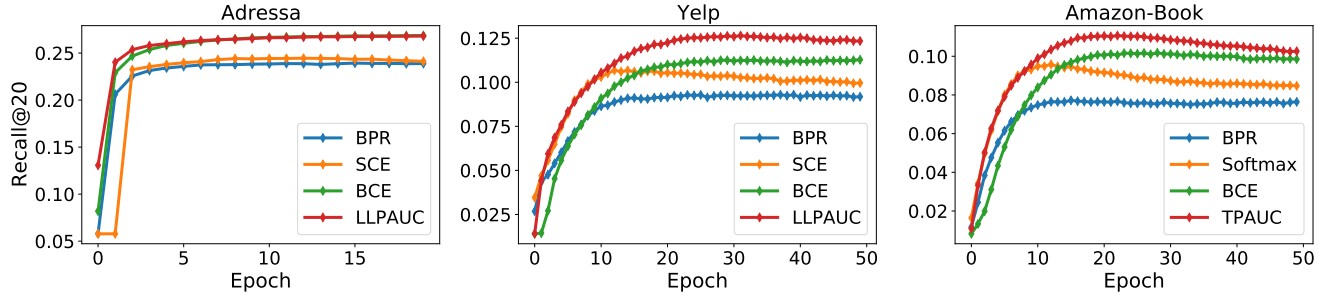

Figure 6: Convergence of different models on three datasets under clean training setting.

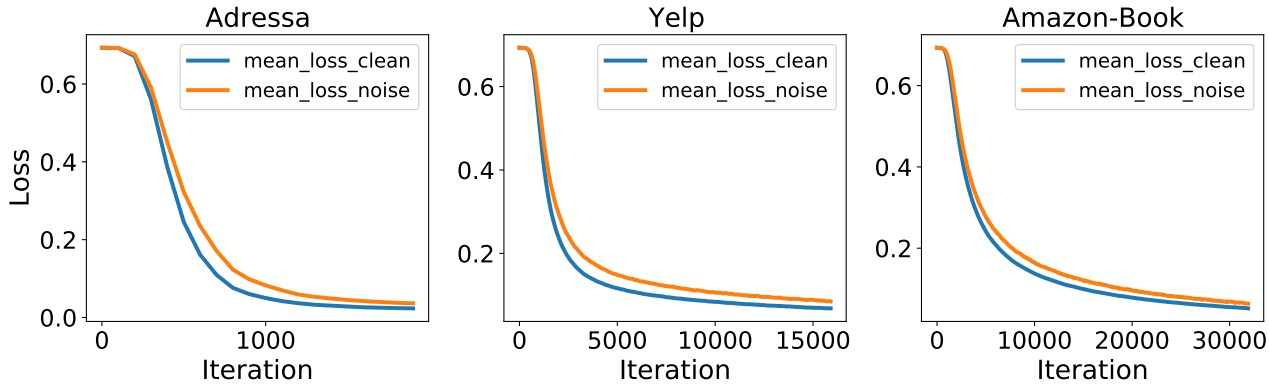

Figure 7: Under noise training setting, the mean loss of noisy interactions and clean interactions *w.r.t* iterations on three datasets.

