# OpenReview forum: "Lower-Left Partial AUC: An Effective and Efficient Optimization Metric for Recommendation"
_ACM.org/TheWebConf/2024/Conference — TheWebConf24_

### Official Review · Reviewer_oikj · 2023-11-22

**Novelty:** 4
**Technical Quality:** 6

**Review:**

summary:

The authors propose LLPAUC which has a higher correlation of ranking metrics (such as NDCG or Recall) than AUC and is computationally efficient like AUC. Then the authors propose the relaxation of LLPAUC for optimization under the deep learning framework. By optimizing the surrogate loss of LLPAUC, the model would get better top-k ranking performance than AUC and the training efficiency is the same as the point-wise loss functions. The author also claims that another advantage of LLPAUC is that its robustness for noisy data. The authors conduct experiments both in clean training and noisy training settings.

strength:

1. Effective and Efficient optimization for recommendation systems is an important and valuable research topic.
2. The proposed LLPAUC is a novel metric which is closer to top-k ranking metrics than AUC.

weakness:

1. Although the computational efficiency of pairwise is slightly lower than that of pointwise, the computational complexity the pairwise methods bring is not a real-world problem. The advanced pairwise or listwise learning-to-rank methods are often used to optimize the model in real-world recommendation and advertising systems. However, the experimental part lacks a comparison with these methods.

**Questions:**

1. The results of the mainstream LTR methods are curious, such as the Lambda Framework.
Xuanhui Wang Cheng Li Nadav Golbandi Mike Bendersky Marc Najork Proceedings of The 27th ACM International Conference on Information and Knowledge Management (CIKM '18), ACM (2018), pp. 1313-1322

**Reviewer Confidence:**

3: The reviewer is confident but not certain that the evaluation is correct

**Scope:**

3: The work is somewhat relevant to the Web and to the track, and is of narrow interest to a sub-community

---

### Official Review · Reviewer_dGPW · 2023-11-24

**Novelty:** 5
**Technical Quality:** 6

**Review:**

This paper investigates Lower-Lefft Partial AUC (LLPAUC) as an adaption of the AUC metric which introduces constraints on the upper bounds for true and false positive rates (TPR and FPR). The aim of this metric is to be computationally as efficient as AUC, but better correlate with Top-K ranking metrics while being robust to noisy user feedback. The paper presents exhaustive experiments across three different datasets for two models.

# Strengths

- The work effectively extends beyond preliminary approaches into that direction, e.g. with OPAUC.
- The code was actually shared upfront and made available which facilitates reproducing the work.
- Results are reported with standard deviation and show up to be significant and substantial.

# Weaknesses

- L299: should be AUC = LLPAUC(1,1)
## Empirical Analysis
- Missing information on how many runs the reported results are based on.
- Table 3: Highlighting for Amazon, NDCG@20 for LightCGN is not correct

# Formal Comments
Wording, writing improvements, etc. - referring to the respective lines below:
- L114: noisy
- L277: missing whitespaces
- L834: resulting in

**Questions:**

- Can you please elaborate on the number of evaluation runs to put the standard deviation into context?

**Ethics Review Description:**

.

**Reviewer Confidence:**

3: The reviewer is confident but not certain that the evaluation is correct

**Scope:**

4: The work is relevant to the Web and to the track, and is of broad interest to the community

---

### Official Review · Reviewer_Gs2H · 2023-11-29

**Novelty:** 5
**Technical Quality:** 7

**Review:**

The paper introduces a new optimization metric called Lower-Left Partial AUC (LLPAUC). This metric is shown to have a better correlation with Top-K ranking metrics than previous metrics. The authors provide both theoretical and empirical evidence to support their claims. They conducted extensive experiments on several datasets and baselines, which confirmed the effectiveness of their proposed method.

Pros:
- The paper is well-written and technically sound. All the theoretical analyses including proofs are presented either in the main text or Appendix;
- The method is built on top of solid research;
- An optimization procedure is presented for the newly proposed metric;
- The results confirm that research claims on three different datasets and two recommendation models considering six different baselines.

Cons:
- Although I am convinced, that adding more datasets and methods to optimize the proposed metric could provide stronger evidence about the generality of the approach. In particular, other commonly used benchmarks such as Movielens;
- The idea has limited novelty since most of the theoretical building blocks are present in previous works.

**Questions:**

Can you specify which aspect of your approach you consider to be the most unique or innovative?

**Reviewer Confidence:**

4: The reviewer is certain that the evaluation is correct and very familiar with the relevant literature

**Scope:**

4: The work is relevant to the Web and to the track, and is of broad interest to the community

---

### Decision · Program_Chairs · 2024-01-22

**Decision:**

Accept

**Comment:**

This work presents a novel optimization metric, Lower-Left Partial AUC (LLPAUC), demonstrating its superior correlation with Top-K ranking metrics compared to existing metrics. The reviewers mostly concur that the paper represents a substantial contribution to the field. They recommend that the authors incorporate the material discussed into the final version of the paper.